

# Agricultural ammonia emissions in China: reconciling bottom-up and top-down estimates

Lin Zhang[1], Youfan Chen[1], Yuanhong Zhao[1], Daven K. Henze[2], Liye Zhu[3], Yu Song[4], Fabien Paulot[5], Xuejun Liu[6], Yuepeng Pan[7], Binxiang Huang[8],

[1]Laboratory for Climate and Ocean-Atmosphere Sciences, Department of Atmospheric and Oceanic Sciences, School of Physics, Peking University, Beijing 100871, China

[2]Department of Mechanical Engineering, University of Colorado, Boulder, Colorado 80309, USA

[3]Department of Atmospheric and Oceanic Sciences, University of California, Los Angeles, California, 90095, USA

[4]State Key Joint Laboratory of Environmental Simulation and Pollution Control, Department of Environmental Science, Peking University, Beijing, 100871, China

[5]Program in Atmospheric and Oceanic Sciences, Princeton University, Princeton, New Jersey 08540, USA

[6]Key Laboratory of Plant-Soil Interactions of MOE, College of Resources and Environmental Sciences, China Agricultural University, Beijing, 100094, China

[7]State Key Laboratory of Atmospheric Boundary Layer Physics and Atmospheric Chemistry (LAPC), Institute of Atmospheric Physics, Chinese Academy of Sciences, Beijing 100029, China

[8]Department of Agrometeorology, College of Resources and Environmental Sciences, China

Agricultural University, Beijing, 100193, China

*Correspondence to:*

Lin Zhang (zhanglg@pku.edu.cn; Tel: 86-10-62766709; Fax: 86-10-62751094)



## Abstract

Current estimates of agricultural ammonia ($NH_3$) emissions in China differ by more than a factor of 2, hindering our understanding of their environmental consequences. Here we apply both bottom-up statistical and top-down inversion methods to quantify $NH_3$ emissions from agriculture in China for the year 2008. We first assimilate satellite observations of $NH_3$ column concentration from the Tropospheric Emission Spectrometer (TES) using the GEOS-Chem adjoint model to optimize Chinese anthropogenic $NH_3$ emissions at the 1/2°×2/3° horizontal resolution for March-October 2008. Optimized emissions show a strong summer peak with emissions about 50% higher in summer than spring and fall, which is underestimated in current bottom-up $NH_3$ emission estimates. To reconcile the latter with the top-down results, we revisit the processes of agricultural $NH_3$ emissions, and develop an improved bottom-up inventory of Chinese $NH_3$ emissions from fertilizer application and livestock waste at the 1/2°×2/3° resolution. Our bottom-up emission inventory includes more detailed information on crop-specific fertilizer application practices and better accounts for meteorological modulation of $NH_3$ emission factors in China. We find that annual anthropogenic $NH_3$ emissions are 11.7 Tg for 2008 with 5.05 Tg from fertilizer application and 5.31 Tg from livestock waste. The two sources together account for 88% of total anthropogenic $NH_3$ emissions in China. Our bottom-up emission estimates also show a distinct seasonality peaking in summer, consistent with top-down results from the satellite-based inversion. Further evaluations using surface network measurements show that the model driven by our bottom-up emissions well reproduces the observed spatial and seasonal variations of $NH_3$ gas concentrations and ammonium ($NH_4^+$) wet deposition fluxes over China, providing additional credibility to the improvements we have made to our agricultural $NH_3$ emission inventory.



## 1. Introduction

Ammonia (NH$_3$) and its aerosol-phase product ammonium (NH$_4^+$) exert important influences on atmospheric chemistry and biodiversity. They contribute to formation of fine particulate matter (PM) that have adverse effects on air quality and visibility (Park et al., 2004; Lelieveld et al., 2015) and cause a cooling climatic forcing (Martin et al., 2004; Henze et al., 2012). Their deposition to nonagricultural ecosystems can further lead to soil acidification and eutrophication (Stevens et al., 2004; Bowman et al.,

2008). Quantifying these environmental consequences requires accurate knowledge of NH$_3$ sources, which are mainly associated with agricultural farming and livestock production (Bouwman et al., 1997). China, due to its intensive agricultural activities, is one of the largest NH$_3$ emitting countries in the world. However, current estimates of Chinese agricultural NH$_3$ emissions differ by more than a factor of 2 (see Sect. 2). Here we aim to better constrain agricultural NH$_3$ emissions in China using available

NH$_3$ concentration and wet deposition flux measurements interpreted by the GEOS-Chem chemical transport model (CTM) and its adjoint.

As the main alkaline gas in the atmosphere, NH$_3$ reacts with sulfuric acid (H$_2$SO$_4$) and nitric acid (HNO$_3$), which are produced by the oxidation of sulfur dioxide (SO$_2$) and nitrogen oxides (NO$_x$), to

form ammonium sulfate and ammonium nitrate aerosols, respectively. These secondary inorganic aerosols account for 40%-57% of the fine PM concentrations in the eastern China (Yang et al., 2011; Huang et al., 2014). Recent studies also highlighted the possible role of NH$_3$ in neutralizing aerosol pH that can strongly enhance formation of sulfate through heterogeneous oxidation of SO$_2$ (Wang et al., 2016; Cheng et al., 2016; Paulot et al., 2016). All this evidence leads to increasing concerns that the

effectiveness of SO$_2$ and NO$_x$ emission controls on fine PM pollution over China may be undermined by unregulated NH$_3$ emissions (Wang et al., 2013; Fu et al., 2017).

Emissions of NH$_3$ are generally estimated from bottom-up statistical methods or process-based models by considering activity data and emission factors (emissions per unit activity) of all possible sources.

Two most important NH$_3$ sources are application of synthetic fertilizers generated by the Haber-Bosch process (Erisman et al., 2008) and livestock production (volatilization of NH$_3$ from animal excreta).



They together contribute 57% of global $NH_3$ emissions (Bouwman et al., 1997) and 80% in Asia (Streets et al., 2003; Kurokawa et al., 2013). Bottom-up $NH_3$ emission estimates highly depend on the accuracy of activity data and emission factors that require detailed spatial and temporal information on

local agricultural practices and environmental conditions as will be discussed in Sect. 2.

Inverse modeling methods provide top-down emission estimates through optimizing comparisons of model simulations with measurements (Gilliland et al., 2003; 2006; Pinder et al., 2006; Zhu et al., 2013; Paulot et al., 2014). Top-down estimates of $NH_3$ emissions over China have been rare due to limited

concentration or flux measurements of reduced nitrogen ($NH_x$ = gaseous $NH_3$ + aerosol $NH_4^+$). A previous inversion study by Paulot et al. (2014) used $NH_4^+$ wet deposition flux measurements from the Acid Deposition Monitoring Network in East Asia (EANET) that only included two sites in China over the studying period. Satellite observations of atmospheric $NH_3$ concentration are emerging in recent years. These satellite instruments include the Tropospheric Emission Spectrometer (TES) (Beer et al.,

2008; Shephard et al., 2011), the Infrared Atmospheric Sounding Interferometer (IASI) (Clarisse et al., 2009; Van Damme et al., 2015), the Atmospheric Infrared Sounder (AIRS) (Warner et al., 2016; 2017), and the Cross-track Infrared Sounder (CrIS) (Shephard and Cady-Pereira, 2015), providing increasingly rich datasets to understand the spatial and temporal variability of $NH_3$ in the atmosphere.

In this study, we apply TES satellite observations of $NH_3$ column concentration to provide top-down constraints on $NH_3$ emissions in China for the year 2008 using the GEOS-Chem adjoint model at the 1/2°×2/3° horizontal resolution. In order to reconcile with the bottom-up estimates and to better understand inversion results, we construct a new bottom-up inventory of Chinese agricultural $NH_3$ emissions by using more practical fertilizer application rates and timing over different crop categories

and by better considering the seasonal variability of emission factors. We further evaluate the top-down and the improved bottom-up Chinese $NH_3$ emissions using an ensemble of surface measurements of $NH_3$ gas concentration and $NH_4^+$ wet deposition flux.

## 2.  Previous bottom-up estimates of Chinese $NH_3$ emissions



We summarize in Table 1 published bottom-up estimates of Chinese $NH_3$ emissions. Annual $NH_3$ emissions from fertilizer application, livestock waste, human excrement, and other sources (e.g., transportation, waste disposal, industry, etc.) are presented from each inventory. Total anthropogenic $NH_3$ emissions based on the years of 2005-2012 range from 8.4 to 18.3 teragram (Tg) $NH_3$ per annum ($a^{-1}$). The factor of 2 difference is not likely due to the different base years. Analyses of historical $NH_3$

emissions in China show relatively stable or weak increasing trends (less than 3% per year) since 2000 (Xu et al., 2016; Kang et al., 2016), consistent with trends in atmospheric $NH_3$ concentration observed from satellites (Warner et al., 2017; Fu et al. 2017).

Fertilizer application and livestock waste are the two largest $NH_3$ sources, together accounting for more

than 82% of the total anthropogenic emissions over China (Table 1). However, considerable differences exist in their emission totals and relative importance. Estimates of $NH_3$ emissions from fertilizer application in China range from 1.82 Tg $a^{-1}$ in 2004 (Li and Li, 2012) to 9.82 Tg $a^{-1}$ in 2010 (Zhao et al., 2013). All these emission estimates are calculated by multiplying fertilizer use amounts with corresponding volatilization rates (emission factors) except for Fu et al. (2015) that considered bi-

directional $NH_3$ fluxes over an agricultural model. Large differences are mainly due to uncertainties in $NH_3$ emission factors that are highly sensitive to fertilizer types, local soil and meteorological properties (Bouwman et al., 2002; Søgaard et al., 2002). $NH_3$ emissions from livestock waste also range from 2.88 to 8.82 Tg $a^{-1}$. An important uncertainty is also attributed to emission factors from livestock waste that heavily relied on European-based measurements in earlier estimates (Streets et al., 2003). Moreover,

some estimates, e.g., Yan et al. (2013) (2.48 Tg $a^{-1}$ for 1995) and Xu et al. (2016) (3.8 Tg $a^{-1}$ for 2008), only accounted for livestock manure spreading to cropland and omitted contributions from animal housing and manure storage.

$NH_3$ from human, including latrines and human perspiration and respiration, is another source with

considerable differences (0.12-1.81 Tg $a^{-1}$). The major component of this source is from rural excrement stored in roughly constructed latrines without sewage service with uncertainties in estimates of rural population and associated $NH_3$ emission factor. Other sources, such as agricultural burning,



transportation, waste disposal, and industry also contribute 0.14-2.8 Tg $NH_3$ $a^{-1}$, depending on inclusions of different source sectors in the emission inventories. For example, Dong et al. (2010) (0.14

Tg $a^{-1}$) only estimated $NH_3$ emitted from chemical industry. These sources are relatively small compared to agricultural sources of fertilizer application and livestock waste at the national scale, however, recent studies show that fuel combustion, (Pan et al., 2016), transportation (Chang et al., 2016; Sun et al., 2017), or local green space (Teng et al., 2017) can be dominant sources of $NH_3$ in the urban atmosphere.


We find substantial differences in spatial and seasonal variations of $NH_3$ emissions among the inventories. Figure 1 compares spatial distributions of anthropogenic $NH_3$ emissions in China from three commonly used inventories: the Regional Emission in Asia (REAS-v2) inventory (Kurokawa et al., 2013), the inventory of Huang et al. (2012), and the Emission Database for Global Atmospheric

Research (EDGAR) (Olivier and Berdowski, 2001). Although they all show higher $NH_3$ emission rates in the east than the west with the highest emissions occurring over North China, there are distinct regional differences of 50-200%. In particular, EDGAR shows much more evenly distributed $NH_3$ emission rates spreading over China than REAS-v2 and Huang et al. (2012).

Figure 1 also shows seasonal variations in these Chinese $NH_3$ emissions. Some inventories such as EDGAR and REAS-v2 do not consider the seasonality of $NH_3$ emissions due to a lack of reliable information. More recent estimates account for information on the timing of fertilizer application and influences of meteorology on $NH_3$ emission factors. We can see that Huang et al. (2012) suggests a weak summer peak in Chinese $NH_3$ emissions, while the MASAGE inventory (Paulot et al., 2014)

indicates largest emissions in April and July. All these discrepancies as discussed above emphasize the needs to improve our understanding of Chinese $NH_3$ emissions in light of measurements of $NH_3$ gas concentration and deposition flux.

## 3. Model description

**3.1. The GEOS-Chem model**



Here we will use the GEOS-Chem CTM and its adjoint to simulate the sources and sinks of $NH_3$ over

China. GEOS-Chem is a global 3-D tropospheric chemistry model (http://geos-chem.org) driven by

assimilated meteorological data from the Goddard Earth Observing System (GEOS) of the NASA

Global Modeling and Assimilation Office (GMAO). The GEOS-5 meteorological data has a horizontal

resolution of 1/2° latitude × 2/3° longitude and a temporal resolution of 3 hours (1 hours for surface

variables). We apply here a one-way nested-grid version of GEOS-Chem with the native 1/2°×2/3°

horizontal resolution over East Asia (70°E-140°E, 15°N-55°N) and 2°×2.5° over the rest of the world

(Wang et al., 2004; Chen et al., 2009).

The model simulates a detailed tropospheric ozone-$NO_x$-hydrocarbon-aerosol chemistry as described by

Park et al. (2004) and Mao et al. (2010). $NH_3$ in the atmosphere is partitioned to gas and aerosol phases

based on the Regional Particulate Model Aerosol Reacting System (RPMARES) thermodynamic

equilibrium model (Binkowski and Roselle, 2003). $NH_3$ prefers to combine with $H_2SO_4$ to form

ammonium bisulfate and ammonium sulfate, and excessive $NH_3$ can react with $HNO_3$ to form

ammonium nitrate. GEOS-Chem simulations of secondary inorganic aerosols (ammonium, sulfate, and

nitrate) over China have been validated by Wang et al. (2013) and Li et al. (2016) recently; both show

high sensitivity of simulated nitrate concentrations to $NH_3$ emissions.

The model wet deposition scheme is described by Liu et al. (2001) with updates from Amos et al. (2012)

for soluble gases and Wang et al. (2011) for aerosols. It includes convective updraft scavenging as well

as large-scale precipitation rainout and washout. Uptake of gaseous $NH_3$ is estimated following Henry's

law in warm clouds ($T > 268$ K), using a retention efficiency of 0.05 in mixed clouds ($248 < T < 268$ K),

and zero efficiency in cold clouds ($T < 248$ K), while aerosol $NH_4^+$ is fully incorporated in all clouds.

Dry deposition calculation follows a standard resistance-in-series model as described by (Wesely, 1989)

for gases and Zhang et al. (2001) for aerosols.

We use the EDGAR global anthropogenic emissions overwritten by regional emission inventories

including the US EPA 2005 National Emissions Inventory (NEI-2005), the European Monitoring and



Evaluation Programme (EMEP) emissions, and the Canada Criteria Air Contaminants (CAC) inventory.

Asian anthropogenic emissions are from Zhang et al. (2009) except for $NH_3$ as described below. These global and regional inventories are scaled to the simulation year of 2008 using the energy statistics as implemented by van Donkelaar et al. (2008). For the prior $NH_3$ emissions, we use the REAS-v2 emission inventory that does not consider any seasonal variation (Kurokawa et al., 2013) so that the inverted emission seasonality is solely from satellite observations. We also follow Zhu et al. (2013) and

increase $NH_3$ emissions from fertilizer use and livestock by 90% in the daytime and reduce them by 90% at night to account for their diurnal variability. Natural sources of $NH_3$ and $NO_x$ from biomass burning, soil and, lightning follow the settings of Zhao et al. (2015), and are relatively small over China (0.56 Tg $NH_3$ $a^{-1}$; Zhao et al., 2017).

## 3.2. The GEOS-Chem adjoint

We use the adjoint of GEOS-Chem to optimize Chinese $NH_3$ emissions through assimilation of TES $NH_3$ column measurements as will be discussed in the next section. The model adjoint is first described by Henze et al. (2007) for aerosols and Kopacz et al. (2009) for carbon monoxide. It has been highly validated and applied in studies to analyze aerosol sensitivities and constrain aerosol sources in the US

(Henze et al., 2009; Zhu et al., 2013; Mao et al., 2015) and China (Kharol et al., 2013; Zhang et al., 2015; 2016; Qu et al., 2017).

The emission optimization is conducted by minimizing the cost function ($J$), defined as:

$$J(\mathbf{x}) = (Y(\mathbf{x}) - \mathbf{y_{obs}})^T \mathbf{S_e}^{-1} (Y(\mathbf{x}) - \mathbf{y_{obs}}) + (\sigma - 1)^T \mathbf{S_a}^{-1} (\sigma - 1) \qquad (1)$$

where $\mathbf{y_{obs}}$ is the vector of satellite observations, $\mathbf{x}$ is the vector of $NH_3$ emissions in the model, $Y(\mathbf{x})$ represents simulated $NH_3$ concentration for comparison with $\mathbf{y_{obs}}$, $\mathbf{x_a}$ is the vector of a priori emissions, $\sigma$ is the vector of scaling factors ($\mathbf{x}/\mathbf{x_a}$) for optimizing, and $\mathbf{S_a}$ and $\mathbf{S_e}$ are the a priori and observational error covariance matrices, respectively. Zhu et al. (2013) has previously applied the adjoint inverse model assimilating TES $NH_3$ data to constrain US $NH_3$ emissions, and Paulot et al. (2014) used wet

$NH_x$ deposition data to constrain East Asian $NH_3$ emissions, both at a coarser 2°×2.5° model resolution.



The adjoint model computes the gradient of the cost function ($\nabla_{\mathbf{x}} J$) numerically, and applies the quasi-Newton L-BFGS-B algorithm (Byrd et al., 1995) to minimize the cost function iteratively. It usually takes about 15 iterations to reach the convergence, identified as the iteration when the cost function

decreases by less than 2% relative to the prior one. To lower the computational expenses, we follow the approach of Paulot et al. (2014) and use an offline $NH_x$ ($NH_3 + NH_4^+$) simulation for the iterative adjoint inversions. The offline $NH_x$ simulation calculates the physical and chemical transformation of $NH_3$ driven by hourly simulated sulfate and total nitrate ($HNO_3 + NO_3^-$) concentrations archived from the standard simulation (Sec. 3.1). This approach would induce errors by not accounting for changes in

total nitrate concentrations when $NH_3$ emissions change (Paulot et al., 2014). We find that a 30% increase of Asian $NH_3$ emissions would increase the total nitrate concentration by about 10%, but deviations of $NH_3$ concentrations in the offline simulation from the standard simulation due to the $NH_3$ emission change are less than 3% over China.

**4. Adjoint inversion of Chinese $NH_3$ emissions with satellite observations**

We use satellite observations of $NH_3$ column concentration over China retrieved from TES, a high-spectral resolution Fourier transform infrared spectrometer aboard the NASA Aura satellite launched in July 2004 (Beer et al., 2006). TES observations have a spatial resolution of 5 km × 8 km at nadir with a local crossing time of 01:30 and 13:30 and global coverage achieved every 16 days (Beer et al., 2008;

Shephard et al., 2011). TES retrievals of atmospheric $NH_3$ concentration are estimated by the optimal estimation method as described in Shephard et al. (2011). We filter the TES $NH_3$ retrievals by only using daytime observations with degree of freedom for signal (DOFS) greater than 0.1. We also correct the positive biases (0.04-1.0 ppbv in the lower troposphere depending on the a priori profile type used in the retrieval) in TES $NH_3$ retrievals following Zhu et al. (2013). Available TES observations for

assimilation then become very spatially sparse for a single month. We assemble TES observations over the years of 2005-2010 for better spatial data coverage. AIRS $NH_3$ observations over 2002-2015 show weak increasing trends (2.27% per year) over Chinese agricultural areas.



Figure 2 shows TES observed $NH_3$ column concentrations with a footprint size of 5 km × 8 km from
March to October. We do not analyze the late fall and winter months (November-February) as the valid
TES observations become very limited, which hinders a reliable emission inversion in those months. As
can be seen in Figure 2, the largest $NH_3$ column concentrations are observed over North China
reflecting intensive agricultural activities over this area. High $NH_3$ concentrations likely related to
animal grazing are also observed over Xinjiang province in Northwest China. TES observations show a
strong seasonality with the national averaged $NH_3$ column concentration a factor two higher in summer
than spring, similar to other satellite $NH_3$ observations retrieved from AIRS (Warner et al., 2017) and
IASI (Van Damme et al., 2015).

We now assimilate TES observed $NH_3$ columns into the model through minimizing the cost function
defined by Eq. (1). The emission optimization is conducted for each month of March-October 2008.
Model results are sampled along the satellite orbits, and then processed with TES a priori profiles and
averaging kernel matrices as a necessary process for comparing with satellite retrievals based on the
optimal estimation method (Zhang et al., 2010; Zhu et al., 2013). For the error covariance matrices, we
assume the a priori error covariance ($S_a$) to be diagonal and the uncertainties to be 100%. The
observational error covariance ($S_{obs}$) is assumed to be 40% of averaged values of observations and
model results accounting for uncertainties in both observations and the model. We have also conducted
sensitivity inversions by using different $S_a$ (50% and 200%) or $S_e$ (20% and 60%) for the July month.

Figure 3 shows differences between TES observed and model simulated $NH_3$ column concentrations
with both the prior and optimized $NH_3$ emissions over China for April, July, and October. It also shows
the correction ratios of optimized emissions over the prior emissions (REAS v2 in Fig. 1). We can see
that with the prior Chinese $NH_3$ emissions, model results largely underestimate TES observations in
July with a mean bias of -47%, while overestimate observations in April and October by 10%-35%.
Model results with optimized $NH_3$ emissions improve comparison correlation coefficients (from 0.25-
0.55 to 0.49-0.63) and significantly reduce the mean biases (-8%-0%). For these inversions, the cost
functions are generally reduced by 35-45%.



The emission correction ratios reflect seasonally and spatially heterogeneous adjustments. Ratios in July show overall increases over China by factors of 1.5-3 except for some locations over Northeast China. In April and October, there are large decreases (up to 70%) over North China and Central China, while increases in Southeast China. Large emission increases are also shown over Northwest China in April and July. The optimized Chinese anthropogenic $NH_3$ emissions in July (1.90 Tg month$^{-1}$) are 47%-57% higher than April (1.21 Tg month$^{-1}$) and October (1.29 Tg month$^{-1}$). Inversion results for March-October indicate that Chinese $NH_3$ emissions peak in summer (see Fig. 7). Similar to previous work of Mao et al. (2015), we also find that the top-down results can be moderately affected by the selection of a priori and observational error covariance matrices. Sensitivity inversions with different $\mathbf{S}_a$ (50% and 200%) or $\mathbf{S}_e$ (20% and 60%) for July show a range of 1.60-1.93 Tg month$^{-1}$, with higher $\mathbf{S}_a$ and lower $\mathbf{S}_e$ values give higher July emission estimates.

## 5. Improving bottom-up estimates of agricultural $NH_3$ emissions

The top-down estimates presented above show a stronger summer peak in Chinese $NH_3$ emissions than those represented in current bottom-up emission inventories (Fig. 1). Reconciling the discrepancy then requires us to better understand the bottom-up emissions from the underlying processes. Previous studies have shown that $NH_3$ emissions are highly sensitive to the magnitude and timing of fertilizer application as well as variations of meteorology (Søgaard et al., 2002; Gyldenkærne et al., 2005; Paulot et al., 2014), but these factors are neither sufficiently represented nor well evaluated in the Chinese $NH_3$ emission estimates. Here we construct an improved bottom-up Chinese $NH_3$ emission inventory from fertilizer application and livestock waste with the objective to better estimate fertilizer application practices and emission factors. Figure 4 shows the schematic diagram of the bottom-up methodology as will be described in detail below.

### 5.1. $NH_3$ emission from fertilizer application

The MASAGE inventory recently developed by Paulot et al. (2014) provides spatial-resolved and crop-specific estimates of fertilizer application practices over the globe. We follow the methodology of



MASAGE for $NH_3$ from fertilizer application but include detailed refinements for China. $NH_3$

emissions from fertilizer application ($E_{NH_3-F}$) are calculated as the product of synthetic fertilizer

application magnitude ($F$) and corresponding emission factors ($\alpha_F$):

$$E_{NH_3-F} = F \times \alpha_F \qquad (2)$$

### 5.1.1. Fertilizer application magnitude


We estimate fertilizer application amounts for 18 crop categories (including early/late rice,

spring/winter wheat, spring/summer maize, cotton, potato, and others as shown in Figure 5). The

fertilizer application magnitude ($F$) is calculated as:

$$F = \sum_c A_C \times \Psi_c(t) \qquad (3)$$

where $A_c$ is the planting area of crop $c$ and $\psi$ represents fertilizer application rate at time $t$ (day of the

year). We use the EarthStat dataset of crop harvest area (Monfreda et al. 2008; EarthStat, 2015;), which

provides global crop harvest areas and yields at 5min × 5min resolution for the year 2000. Here we

regrid them to the model 1/2°×2/3° resolution, and scale the harvest area for each of the 18 crop

categories to the year 2008 using the province-level data from the National Bureau of Statistics of China

(NBSC, 2015). We use the harvest areas as the crop planting areas except for rice and tobacco. Their

planting areas are about 5% of the harvest areas during the seeding period, and then move to the

transplanting fields till harvest.

Estimating the fertilizer application rate $\Psi_c(t)$ need to consider the planting schedule and fertilizer

application practice for each crop. Paulot et al. (2014) distribute the annual fertilizer application amount

over three stages (at planting, at growth, and after harvest) by assuming crop-specific application ratios.

Here we consider much more detailed fertilizer application practices over China. Each crop requires the

basal fertilizer applied at planting and several top dressing fertilizers during its growth (up to five

application times). We construct tables of the fertilizer dates and rates (Supplemental Table S1 and S2)

for the main crop categories in China based on Liao (1993) and Zhang and Zhang (2012). The timing of

fertilizer application to each crop is based on its planting date or the calendar day (Table S1). We use



the crop planting dates of Sacks et al. (2010). Following Paulot et al. (2014), a Gaussian distribution function (Gyldenkærne et al., 2005) is applied to account for uncertainties and interannual variations of application dates:

$$F(t) = F \times \frac{1}{\sigma_c\sqrt{2\pi}} \times e^{\left(\frac{(t-\mu_c)^2}{-2\sigma_c{}^2}\right)} \tag{4}$$

where $\mu_c$ is the crop-specific fertilizer application date, and $\sigma_c$ is the deviation from the mean planting date estimated from the dataset of Sacks et al. (2010) as summarized in Table S1.

The procedures above allow us to estimate fertilizer application rates at each calendar day for the main
crops listed in Table S1 and S2, and we sum them up at the monthly scale. Fertilizers applied through the injection and broadcast modes are estimated separately. We assume that the first fertilizer application at plant is through injection (for rice and tobacco the first applications at both seeding and transplanting fields), and the rest by broadcast. For other crops, fruits, and vegetables, we use the annual fertilizer use amounts from the International Fertilizer Industry Association (IFA 2013) and NBSC
(2015), and then distribute them spatially using the crop yield data of EarthStat (2015) and apply monthly variations proportional to the number of daylight hours following Park et al. (2004).

Figure 5 shows seasonal variations of fertilizer application to each crop category in China through injection and broadcast, separately. We estimate that 9.2 Tg N a$^{-1}$ fertilizers are used through injection,
and 15.8 Tg N a$^{-1}$ through broadcast. They show strong but different seasonal variations as resulted from variations of application timings for different crop categories. Injected fertilizer uses peak in spring (April and May) and have a second peak in fall mainly due to winter wheat, while broadcast fertilizers maximize in late spring and summer. The annual total fertilizer application is estimated to be 25.0 Tg N, compared with 22.4 Tg N in Huang et al. (2012) for 2006 and the FAO (Food and
Agriculture Organization of the United Nations) estimate of 28.5 Tg N for 2008 (FAOSTAT, 2015).

**5.1.2 Emission factor from fertilizer application**



We estimate emission factors of $NH_3$ from fertilizer application as a function of soil properties and agricultural activity information, and further modulated by meteorological conditions (Paulot et al.,

2014). The emission factor is first calculated as:

$$\alpha_0 = e^{f_{pH} + f_{CEC} + f_{type} + f_{crop} + f_{mode}} \tag{5}$$

where the factors (*f*) represent effects of soil pH, cation exchange capacity (CEC), fertilizer type (e.g., urea, ammonium bicarbonate (ABC), ammonium sulfate (AS), and others), and application mode (broadcast and injection) on $NH_3$ volatilization mainly based on Bouwman et al. (2002) (Supplemental

Table S3). Monthly scalars are then applied to account for the seasonality driven by meteorology:

$$\alpha_F = \alpha_0 \left( e^{0.0223T_i + 0.0419W_i} \right) \Big/ \sum_{j=1}^{12} e^{0.0223T_j + 0.0419W_j} \tag{6}$$

where $T_i$ and $W_i$ are 2m (meter) air temperature in °C and 10m wind speed in m s$^{-1}$ for month $i$, respectively (Søgaard et al., 2002; Gyldenkærne et al., 2005).

We use the gridded (0.5°×0.5°) soil pH data from the University of Wisconsin Nelson Institute Center for Sustainability and the Global Environment (SAGE, 2015), and the soil CEC data (0.5°×0.5°) from the ISRIC-World Soil Information (ISRIC WSI, 2015) of the World Data Center for Soils. Crop categories and application modes follow the calculation of fertilizer application rates as described above. Percentages of different fertilizer types applied to cropland are estimated based on the statistics

from Zhang and Zhang (2012).

We have collected an ensemble of published measurements of $NH_3$ emission factors (volatilization rates) from fertilizer application that cover different regions of China and consider different fertilizer types and application modes, as summarized in Table S4. We compare these measurements with

corresponding emission factors derived by Eq. (5). This comparison does not include meteorological effects (Eq. (6)) due to a lack of relevant measurements for the published emission factors. As shown in Figure 6, calculated and measured values are in good agreement with a correlation coefficient of 0.80 and a mean bias of 10%, supporting calculations of $NH_3$ emission factors using Eq. (5) for fertilizer application practices in China.




## 5.2 Livestock waste

The Chinese NH$_3$ emissions from livestock waste are commonly derived as the product of livestock population and emission factors (Streets et al., 2003; Paulot et al., 2014). Here we follow Huang et al. (2012) and Kang et al. (2016) that adopted a more process-based mass-flow approach by considering

the transformation of nitrogen in animal husbandry. As shown in Figure 4, a pool of total ammoniacal nitrogen (TAN) as input to manure management is estimated by animal excreta from three main raising systems (free-range, intensive, and grazing). We use the gridded livestock (e.g., pork, beef, dairy, sheep, poultry, etc.) population from the Gridded Livestock of the World (GLW, 2015), and then adjust them to match the province-level annual records of NBSC (2015) for 2008. The parameters of annual TAN

excretion per animal considering both urine and feces and their nitrogen contents for each livestock category are given in Huang et al. (2012).

We estimate the content of TAN produced outdoors and in house separately by assuming percentages of time spending outside and inside buildings for each livestock category (Huang et al. 2012). The outdoor

TAN is directly deposited in the open air, while the indoor TAN can flow through the stages of housing, storage, and spreading to cropland as basal fertilizer with depletion of TAN from processes such as immobilization and leaching at each stage. NH$_3$ emissions from livestock are calculated as the product of TAN at the four stages and corresponding emission factors. We use the emission factors of Huang et al. (2012) and further account for the meteorological influences as represented by Eq. (6). We consider

both air temperature and wind speed for outdoor NH$_3$ emissions, while only account for air temperature for indoor emissions. Monthly emission factors are calculated using the GEOS-5 assimilated meteorological data at the model 1/2°×2/3° resolution.

## 5.3 Improved emissions and evaluation with surface measurements

Figure 7 presents our improved bottom-up estimates of NH$_3$ emissions from fertilizer application, livestock waste, anthropogenic totals, and their seasonal variations. We adopt estimates of other NH$_3$ sources (agricultural burning, chemical industry, transportation, and waste disposal) from Huang et al.



(2012) for calculating the total anthropogenic NH$_3$ emissions. Our bottom-up estimates show Chinese NH$_3$ emissions of 5.05 Tg a$^{-1}$ from fertilizer application and 5.31 Tg a$^{-1}$ from livestock, and reach 11.7

Tg a$^{-1}$ with addition of other anthropogenic sources. High NH$_3$ emission rates occur over the North China Plain (over 80 kg ha$^{-1}$ a$^{-1}$ in parts of Hebei and Henan provinces) and the Sichuan basin. Zhang et al. (2010) also reported similar high NH$_3$ emission (with highest value up to 198 kg ha$^{-1}$ a$^{-1}$) in the North China Plain in the year 2004. These spatial features are overall comparable to Huang et al. (2012) and REAS v2 as shown in Figure 1. The total anthropogenic emission estimate of 11.7 Tg a$^{-1}$ is in the

middle of previous bottom-up estimates as summarized in Table 1, however, our NH$_3$ emissions show much more distinct seasonal variations with emissions a factor of 3 higher in summer than winter. As shown in Figure 7, this strong seasonality is consistent with the adjoint optimized emission totals for March-October (50% higher in summer than spring and fall) considering uncertainties in the inversion results.


For further and independent evaluation of these NH$_3$ emissions, we use an ensemble of surface measurements of NH$_3$ gas concentration and NH$_4^+$ wet deposition flux compiled by Zhao et al. (2017). The dataset includes monthly averages from a nationwide measurement network over China for 2011-2012 (Xu et al., 2015), ten sites in the North China Plain monitored by the Chinese Academy of

Sciences for 2008-2010 (Pan et al., 2012), and two sites in China from EANET (2015). It should be noted that these measurements are compared with simulated results for 2008, inducing uncertainties from interannual variations.

We show in Figure 8 comparisons of NH$_4^+$ wet deposition flux measurements with model results using

improved bottom-up, prior (REAS v2), and optimized Chinese NH$_3$ emissions. Measured NH$_4^+$ wet deposition fluxes indicate a strong peak in summer with a national averaged monthly flux of 2.8 ± 1.6 kg ha$^{-1}$ month$^{-1}$ and a minimum in winter (0.4 ± 0.3 kg ha$^{-1}$ month$^{-1}$). Part of this seasonality is driven by heavier precipitation in summer as model results with the prior REAS v2 emissions also capture some of the summer vs. winter deposition flux differences. However, model results with the prior

REAS v2 still show underestimates of the flux measurements in summer and overestimates in winter



and spring, while the adjoint optimized $NH_3$ emissions reduce the model biases (e.g., mean biases reduced from 18% to 9% in spring and from -18% to 11% in summer). We can see that model results with our improved bottom-up $NH_3$ emissions well reproduce the spatial and seasonal variations in measured $NH_4^+$ wet deposition fluxes (correlation coefficients $r = 0.41$-$0.70$).


Figure 9 shows comparisons for the surface $NH_3$ gas concentration. Comparing model results with these surface concentration measurements needs to address the inconsistency in the altitude they represent. The lowest model layer centered at 70 m above surface, while all these Chinese surface sites used here measure at 3 m above surface. Zhang et al. (2012) previously quantified the vertical gradient of $HNO_3$

concentrations in the lowest model layer based on the dry deposition resistance-in-series formulation and the Monin-Obukhov similarity theorem. Here we follow the same approach to estimate the 3m/70m gradient of $NH_3$ concentrations driven by net $NH_x$ flux in each grid. Implied $NH_3$ concentrations at 3 m are on average 20%-30% higher than those at 70 m. As a result, model results with our improved bottom-up emissions show better agreement (with seasonal mean biases within ±20%) with surface $NH_3$

concentration measurements than REAS v2 and optimized emissions. The relatively low correlation coefficients (0.14-0.39) may reflect difficulties for the model to fully capture the heterogeneity in $NH_3$ concentration due to its short lifetime, uncertainties gaseous $NH_3$ and aerosol $NH_4^+$ partitioning, and also interannual variations in $NH_3$ measurements.

**6. Conclusions**

In summary, we have applied both bottom-up and top-down methods to better understand agricultural $NH_3$ emissions in China. A review of recent bottom-up estimates of Chinese $NH_3$ emissions shows substantial differences not only in annual estimates of $NH_3$ emissions from fertilizer application and livestock waste, but also in their spatial and seasonal variations. The large differences mainly reflect

limited information on fertilizer application practices and uncertainties in emission factors of $NH_3$ from agricultural activities.



We conduct top-down estimates of $NH_3$ emissions in China by assimilating TES satellite observations of $NH_3$ column concentration with the GEOS-Chem adjoint model for March-October 2008. The

465 optimized Chinese $NH_3$ emissions show a strong summer peak that is generally underestimated in current bottom-up emission estimates. Optimized monthly emissions in July are 1.90 Tg (1.60-1.93 Tg considering different configurations of error covariance), ~50% higher than these in April (1.21 Tg) and October (1.29 Tg).

470 To interpret the top-down emission estimates, we revisit the bottom-up estimate of agricultural $NH_3$ emissions aiming to better estimate the fertilizer application practices and $NH_3$ emission factors in China. We improve the emission inventory of Paulot et al. (2014) for $NH_3$ from fertilizer application with more realistic estimates of fertilizer use magnitudes and growth schedules of main crop categories in China. Emission factors of $NH_3$ from fertilizer application are calculated as a function of fertilizer

475 type, application mode, soil property, and meteorological condition. Our validation of calculated values with an ensemble of published emission factor measurements shows a good agreement. For $NH_3$ emissions from livestock waste, we follow the mass-flow approach of Huang et al. (2012) and further account for meteorological influences (air temperature and wind speed) on $NH_3$ emission factors.

480 We find in our improved bottom-up inventory for the year 2008 that annual Chinese $NH_3$ emissions are 5.05 Tg $a^{-1}$ from fertilizer application and 5.31 Tg $a^{-1}$ from livestock waste. Addition of other anthropogenic $NH_3$ sources from Huang et al. (2012) (1.3 Tg $a^{-1}$) suggests annual anthropogenic emissions of 11.7 Tg $a^{-1}$. The improved bottom-up estimates of Chinese anthropogenic $NH_3$ emissions now display a strong seasonality with emissions in summer ~50% higher than spring and a factor of 3

485 higher than winter, similar to the seasonality in the top-down emission estimates.

We further evaluate the improved bottom-up and top-down Chinese $NH_3$ emissions using available surface measurements of $NH_3$ gas concentration and $NH_4^+$ wet deposition flux. We find that model results with improved bottom-up emissions well reproduce the spatial and seasonal variations in these

490 surface measurements, demonstrating improvements in $NH_3$ emissions resulted from inclusion of



detailed information on fertilizer application practices and seasonal variations of NH$_3$ emission factors. We acknowledge that measurements used in the study (both surface and TES satellite measurements) are still sparse in spatial coverage. Future studies using satellite NH$_3$ observations from AIRS, IASI, and CrIS that have better spatial data coverage will be valuable to constrain the spatial variability of

NH$_3$ emissions.

**Data availability**

The datasets including measurements and model simulations can be accessed from websites listed in the references or by contacting the corresponding author (Lin Zhang; zhanglg@pku.edu.cn). The Chinese

agricultural NH$_3$ emission inventory developed in this study can also be downloaded from the webpage (http://www.phy.pku.edu.cn/~acaq/data/nh3_agr_emis.html).

**Acknowledgements**

This work was funded by the National Key Research and Development Program of China

(2017YFC0210102), China's National Basic Research Program (2014CB441303), and the National Natural Science Foundation of China (41205103, 41425007, and 41405144).

**4 Tables are included in the supplement related to this article.**

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





**Tables and Figures**

**Table 1. Bottom-up estimates of ammonia anthropogenic emissions in China[1]**

| References | Base year | Fertilizer application | Livestock waste | Human | Others[2] | Total |
|---|---|---|---|---|---|---|
| Yan et al. (2003) | 1995 | 4.32 | 2.48[3] | 0.21 | | 7.01 |
| Streets et al. (2003) | 2000 | 6.8 | 5.17 | 1.63 | | 13.6 |
| Li and Li (2012) | 2004 | 1.82 | 8.30 | 1.67 | 0.21 | 12.0 |
| Wang et al. (2009) | 2005 | 4.3 | 8.82 | 0.26 | | 13.38 |
| Zhang et al. (2011) | 2005 | 4.31 | | | | |
| Dong et al. (2010) | 2006 | 8.68 | 6.61 | 0.65 | 0.14 | 16.08 |
| Huang et al. (2012) | 2006 | 3.2 | 5.3 | 0.2 | 1.1 | 9.8 |
| Cao et al. (2010) | 2007 | 3.62 | 9.58 | | 2.8 | 16.0 |
| EDGAR | 2008 | 8.1 | 3.1 | 0.1 | | 11.3 |
| Xu et al. (2016) | 2008 | 3.3 | 3.8[3] | 0.7 | 0.6 | 8.4 |
| Paulot et al. (2014) (MASAGE) | 2008 | 3.6 | 5.8 | 0.8 | | 10.2 |
| Kurokawa et al. (2013) (REAS v2) | 2008 | 9.46 | 2.88 | 1.81 | 0.85 | 15.0 |
| Zhao et al. (2013) | 2010 | 9.82 | 7.36 | 1.12 | | 18.3 |
| Fu et al. (2015) | 2011 | 3 | | | | |
| Kang et al. (2016) | 2012 | 2.8 | 4.99 | 0.12 | 1.71 | 9.62 |
| This study | 2008 | 5.05 | 5.31 | 1.30[4] | | 11.7 |

[1] Emission totals in unit of Tg $NH_3$ a$^{-1}$.

[2] Others include sources from transportation, industry, waste disposal, and agricultural burning.

[3] Only considering $NH_3$ emission from livestock manure spreading to cropland

[4] Emission estimates adopted from Huang et al. (2012).






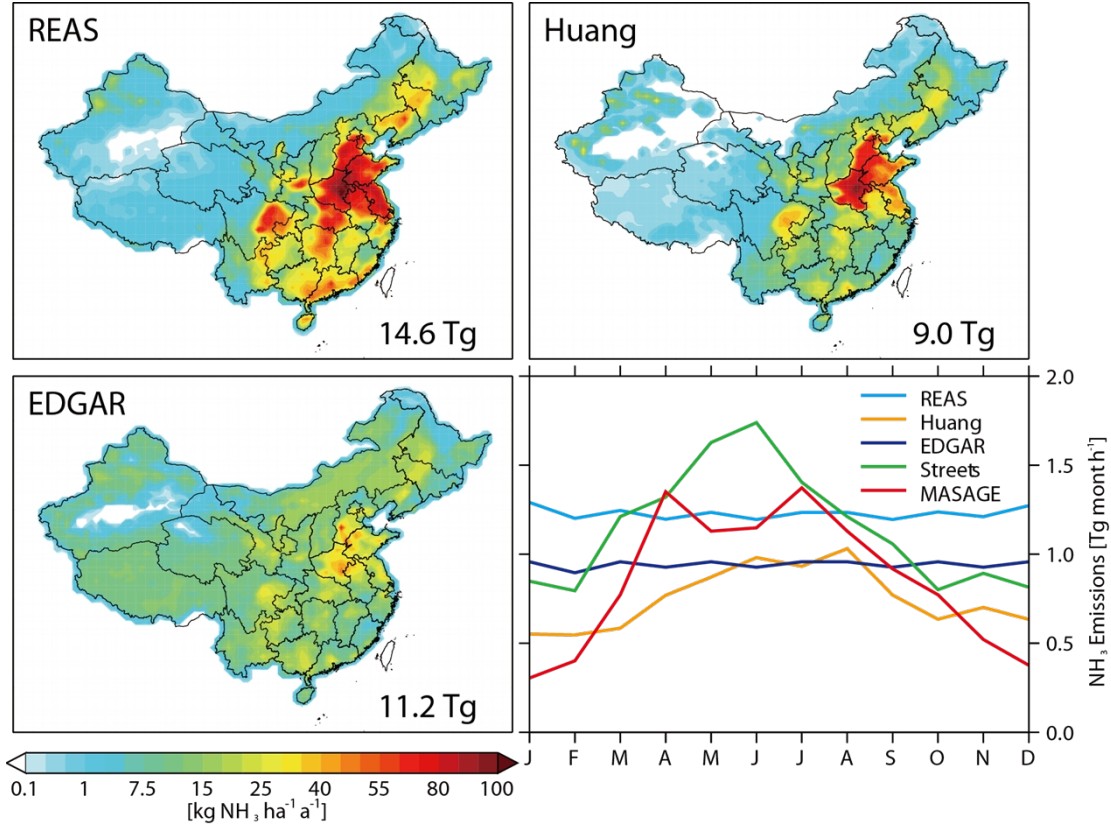

**Figure 1**. Spatial and seasonal variations of anthropogenic NH₃ emissions in China from different

bottom-up inventories. Numbers inset are annual totals of Chinese anthropogenic NH₃ emissions. See

Table 1 for references of the emission inventories.





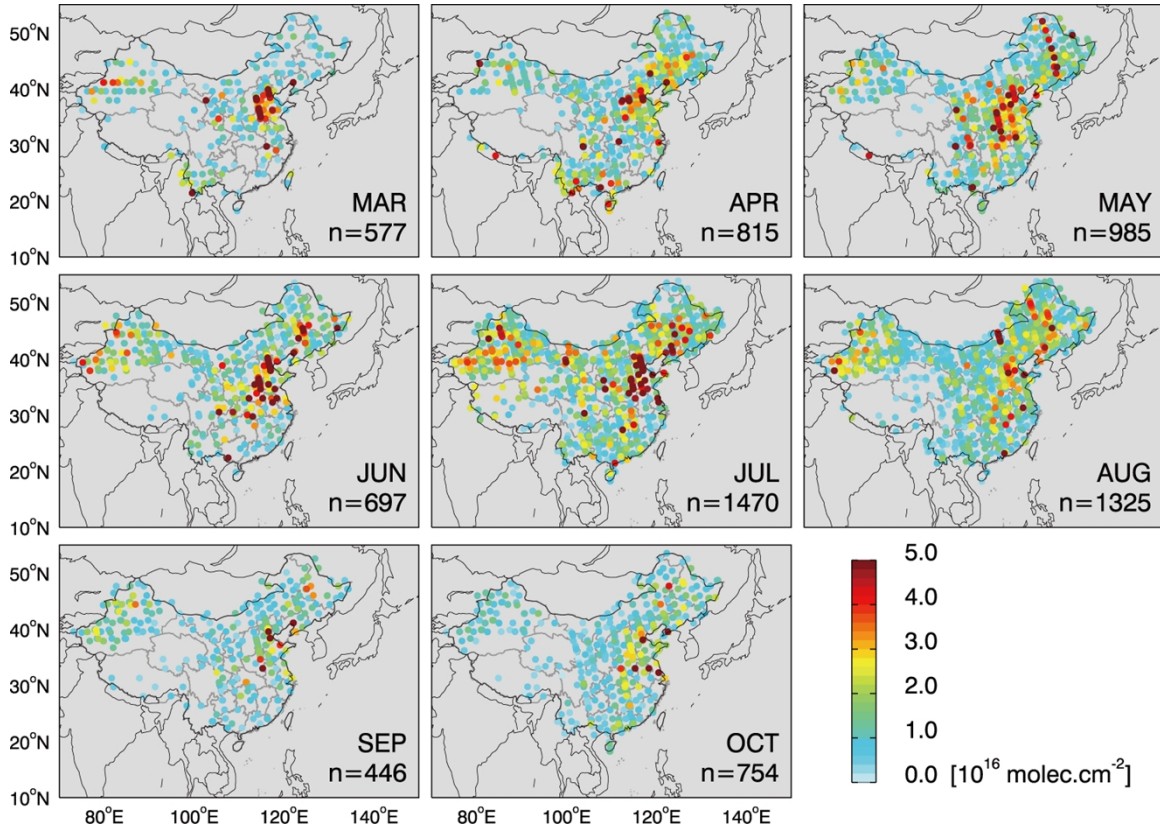

**Figure 2**. TES observations of NH$_3$ column concentration over China from March to October in the years 2005-2010. Each point represents a TES observation with a footprint resolution of 5 km × 8 km. Values inset are number (n) of valid observations for assimilation in each month.




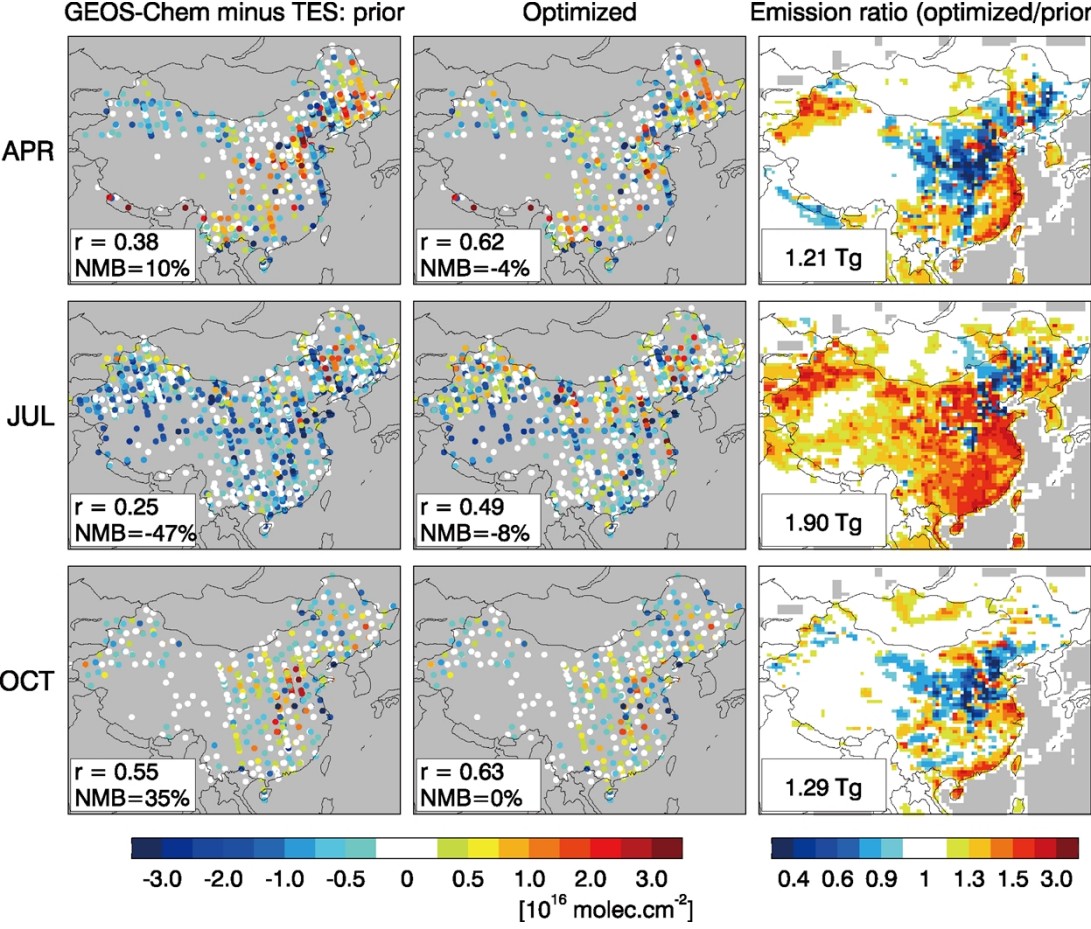

**Figure 3**. Differences between GEOS-Chem simulated and TES observed ammonia column concentrations over China for April, July, and October. The left and middle panels show results from model simulations with prior and optimized ammonia emissions, respectively. Correlation coefficients (*r*) and normalized mean biases (NMB) are shown inset. The right panels show monthly correction ratios relative to the prior emissions with optimized Chinese emission amounts shown inset.






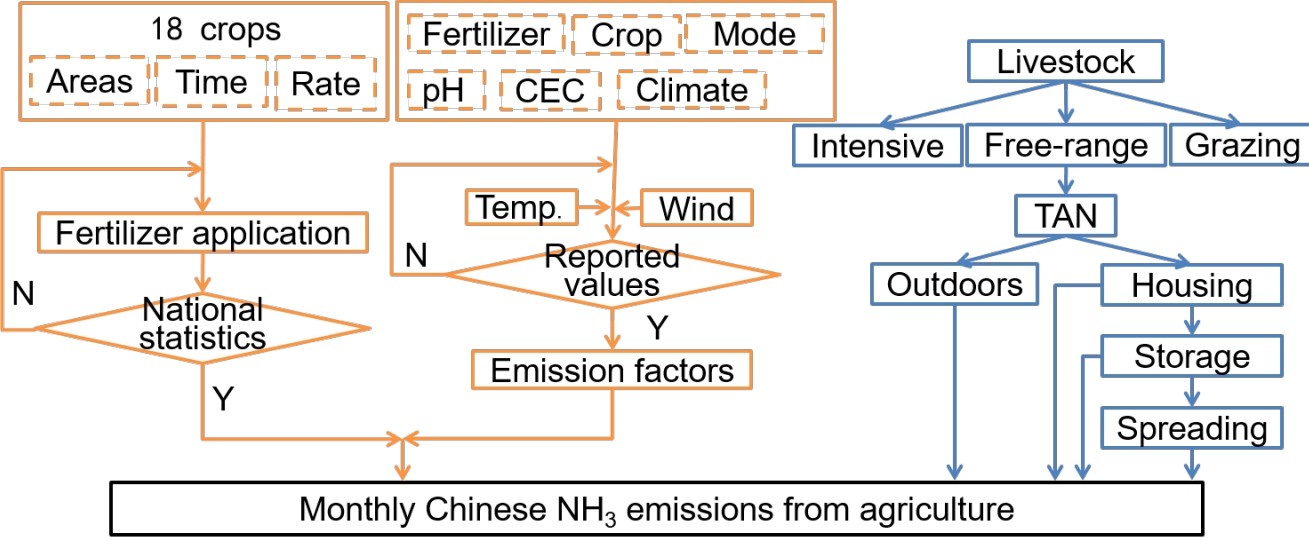

**Figure 4**. Schematic diagram for estimating agricultural NH$_3$ emissions from fertilizer application and livestock waste.






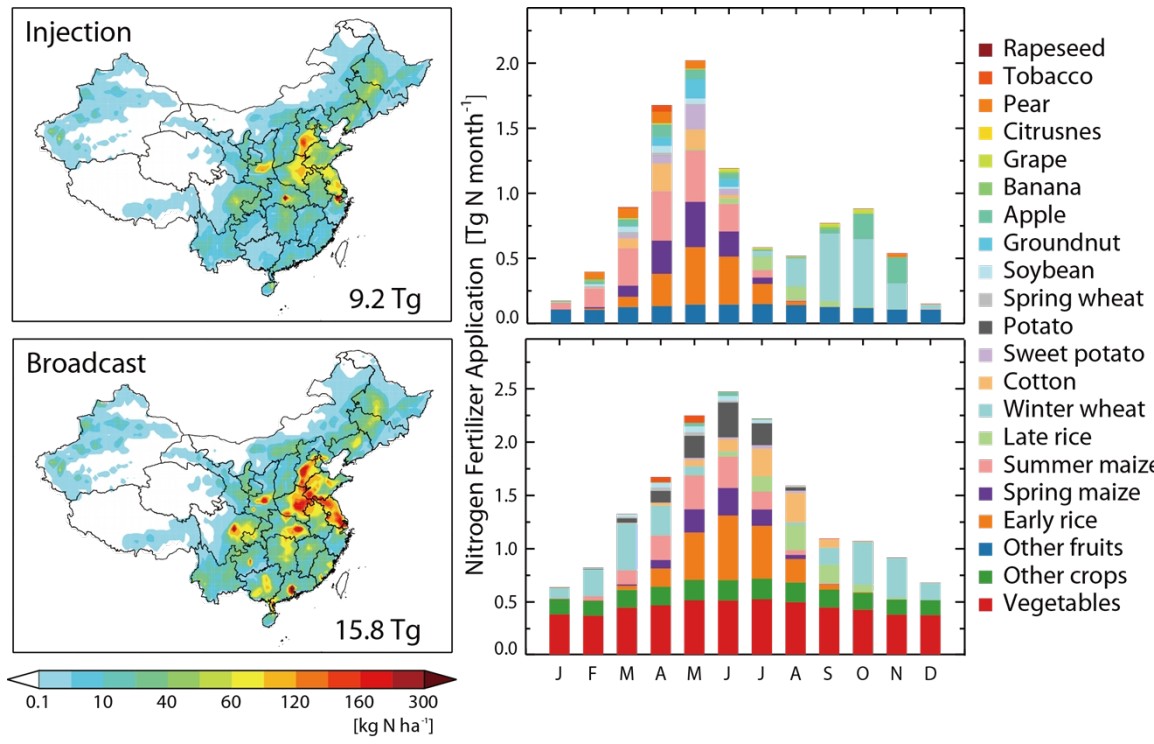

**Figure 5.** Fertilizer application through injection (top panels) and broadcast (bottom panels) techniques

in China for 2008. The left panels show annual total fertilizer application at the 1/2° × 2/3° model

resolution with the annual totals given inset. The right panels show monthly fertilizer application

amounts over China for the 18 crop types.





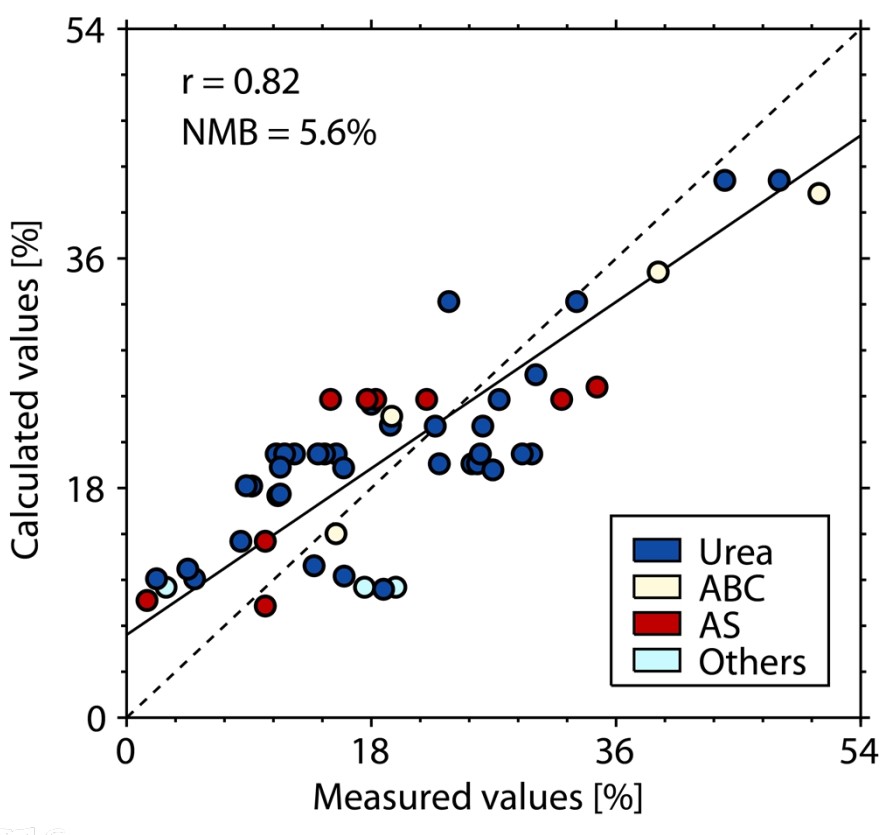


**Figure 6**. Comparison of calculated (using Eq. (5) in the text) and measured $NH_3$ emission factors from application of different fertilizers: urea, ammonium bicarbonate (ABC), ammonium sulfate (AS), and others in China. The correlation coefficient and normalized mean bias (NMB) are shown inset. Measurements of $NH_3$ emission factors are summarized in Supplemental Table S4.






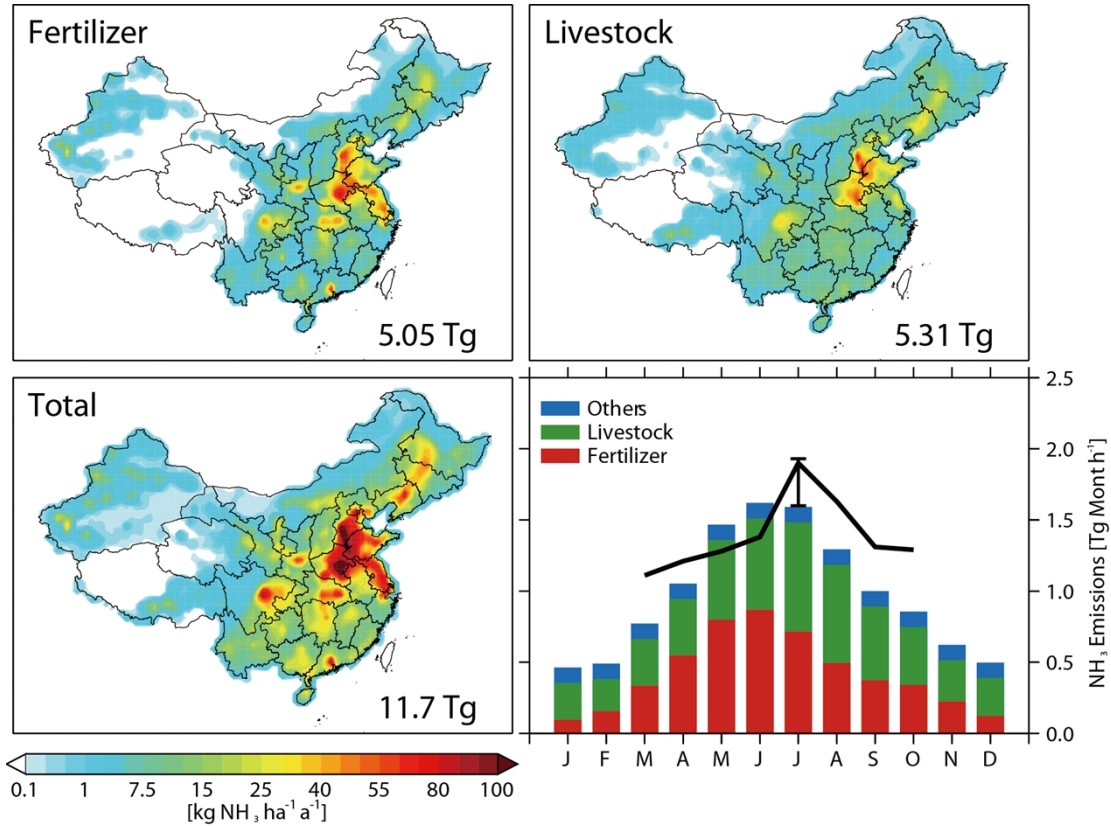

**Figure 7**. Improved bottom-up $NH_3$ emission estimates from fertilizer use (top-left panel), livestock (top-right panel), and total anthropogenic emissions (bottom-left panel) in China. Values inset are emission totals. The bottom-right panel shows seasonal variations in Chinese $NH_3$ emissions from different source categories. The bars represent our bottom-up estimates and the black line shows adjoint optimized anthropogenic totals for March-October. The vertical black line denotes the range of top-down estimates from inversions with different error configurations for the July month as described in the text.





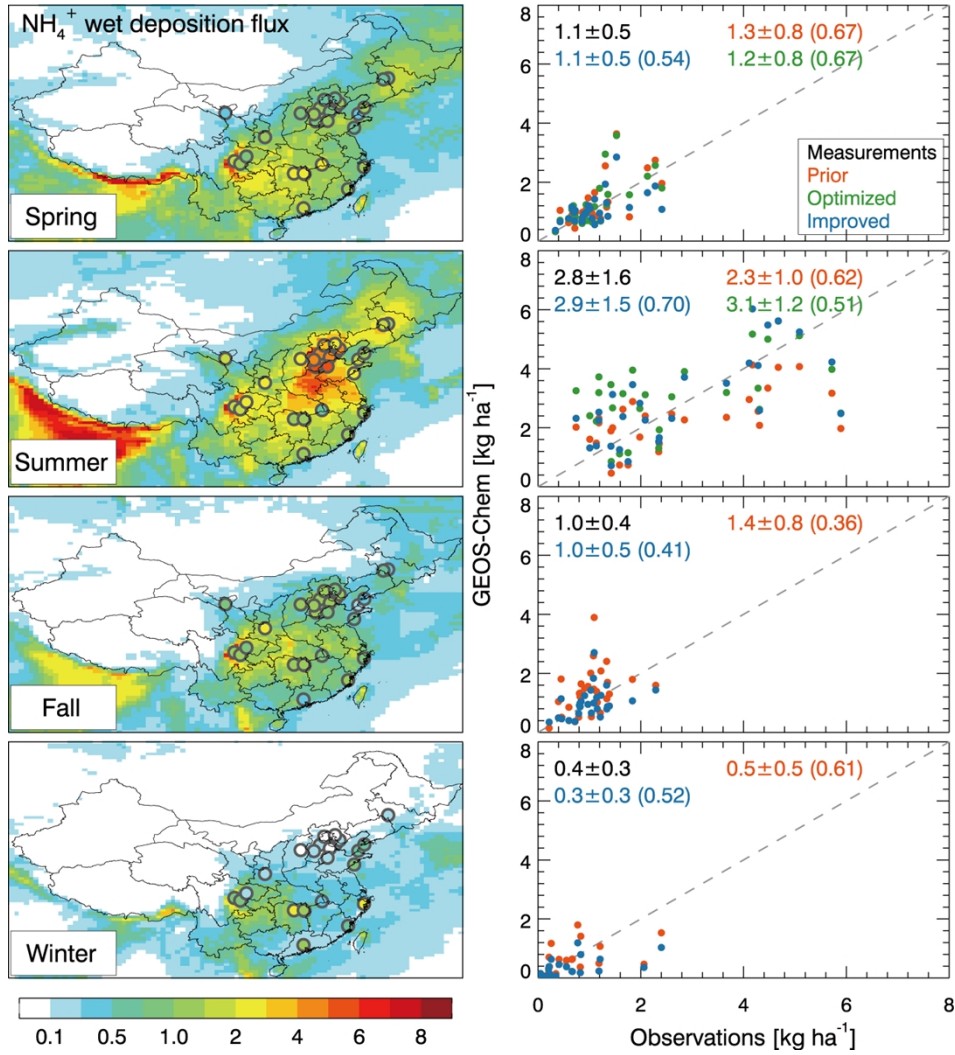

**Figure 8**. Comparison of simulated vs. measured seasonal mean $NH_4^+$ wet deposition fluxes over China. Spatial distributions (left panels) and scatterplots (right panels) are shown. Values inset are seasonal mean deposition fluxes and standard deviations for measurements (black), and model results with prior REAS-v2 (orange), adjoint optimized (green; for spring and summer), and improved bottom-up (blue) $NH_3$ emissions. The correlation coefficients (values in parentheses) and the 1:1 line (dashed line) are also shown.





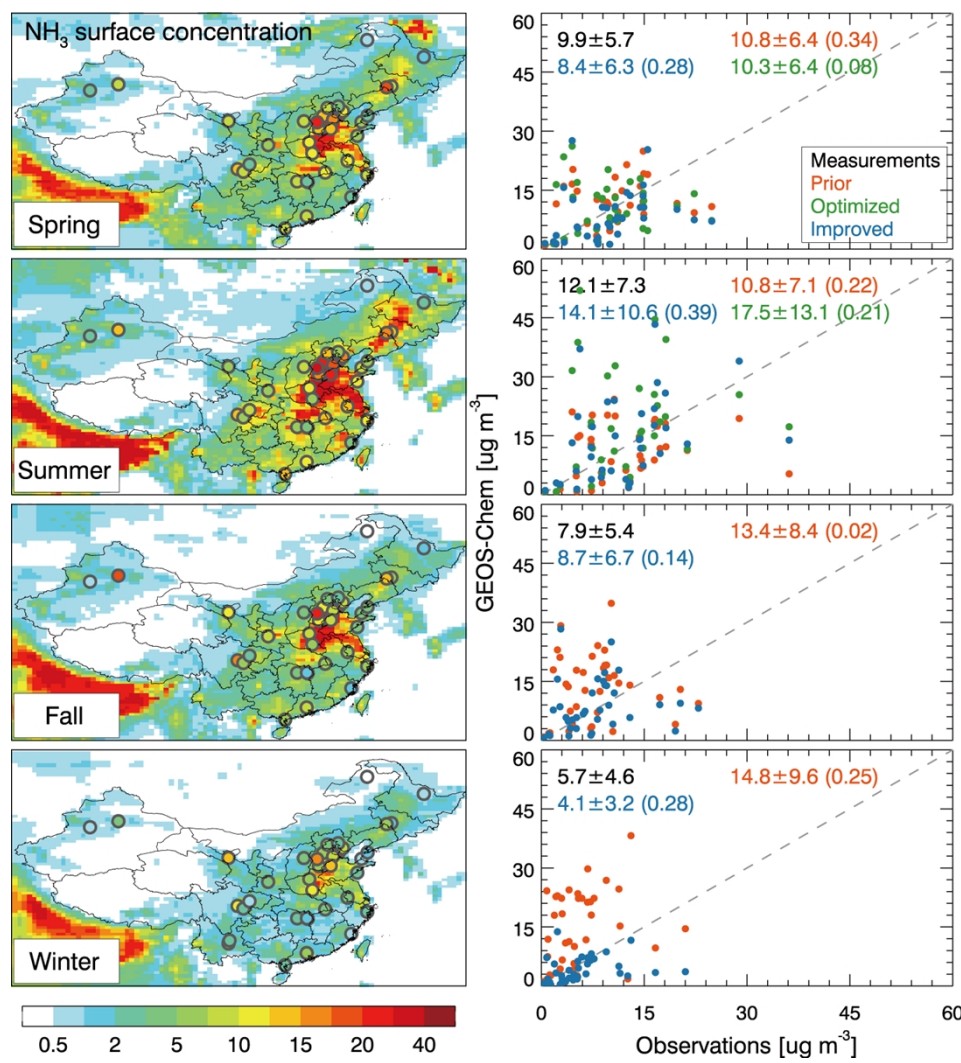

870

**Figure 9**. The same as Figure 8, but for surface NH$_3$ concentration.

875