# Peer review of "Agricultural ammonia emissions in China: reconciling bottom-up and top-down estimates"

_Atmospheric Chemistry and Physics, 2017_

## Referee Comment (RC1) · J. Collett (Referee) · 4 Oct 2017

Zhang et al have done a terrific job using models and observations to improve understanding of ammonia emissions in China. Not only do they do a top-down analysis, using the GEOS-CHEM adjoint constrained by TES column NH3 measurements, to improve the seasonal and spatial variability in NH3 emissions, they then do a very thorough job improving past bottom-up inventories through careful analysis of fertilization practices and animal emissions. Combined, these make for a very strong paper – one of the best I have reviewed in some time.

I recommend the authors attend to a few comments in revising the manuscript:

1. One of the main challenges in accurately simulating ammonia concentrations in

chemical transport models is the treatment of dry deposition. Considerable attention has been paid recently to including more realistic, bi-directional flux parameterizations and this seems to help quite a lot in some regional simulations. Without a bidirectional treatment, NH3 loss rates by dry deposition can be biased high. While I am OK with the authors not including a bidi treatment in their model simulations here, I do think they should add some discussion how its absence might influence their results. This is relevant to the top-down NH3 emissions estimates and to the comparison of model vs. surface concentration and wet deposition estimates.

2. Line 77: I suggest changing "together contribute" to "together are estimated to contribute"

3. Lines 150-157: the authors should discuss the Streets emission inventory here in the text. It is included in the figure and shows the strongest seasonality.

4. Line 173: I suggest changing "NH3 prefers to combine" to "NH3 is thermodynamically favored to combine"

5. Line 182: change "mixed clouds" to "mixed-phase clouds"

6. Lines 182-184: please explain and justify the retention efficiencies chosen for mixed-phase and cold clouds

7. Lines 248-249: are NH3 concentrations possibly also higher here because there are fewer NOx and SO2 emissions to generate acids that tie NH3 up in aerosols?

8. Lines 336-338: How accurate/representative for China are the authors' assumptions here re: frequency of application of injection and broadcast fertilization methods?

9. While the manuscript is generally quite well written, there are several small grammatical errors that should be corrected. The most significant are

a. Line 52: change "have" to "has" and "cause" to "causes"

b. Line 66: change "in the eastern China" to "in eastern China"

c. Line 129: change "human" to "humans"

d. Line 268: change "while overestimate" to "while they overestimate"

e. Line 276: change "increases in" to "increases are noted in"

f. Line 319: change "need to consider" to "requires considering"

g. Line 394: change "spending" to "spent"

h. Line 400: change "while only" to "while we only"

i. Line 442: change "needs to address" to "requires addressing"

j. Line 443: change "layer centered" to "layer is centered"

---

## Referee Comment (RC2) · Anonymous Referee #2 · 11 Oct 2017

This manuscript uses both top-down and bottom-up methods to investigate the spatial and temporal variations of agricultural ammonia emissions in China. The top-down estimates of NH3 emissions, constrained by TES satellite NH3 observations and optimized by GEOS-Chem adjoint model, show a summer peak that is underestimated in current bottom-up emissions inventories. To resolve the seasonal difference, the authors construct a new bottom-up inventory that takes account of seasonal variability in fertilizer application rates and emissions factors. The improved bottom-up inventory is broadly consistent with the top-down inversion results; both are validated by surface concentrations of NH3 and wet deposition fluxes of NH4+. Overall, I think the paper reads well, provides interesting results and deserves publication. I include some minor comments and suggested revisions in the following text.

[Figure]

1. Inverse method. The TES satellite NH3 columns are included in the observation vector, and these measurements are the basis for deriving seasonal variations in inverted NH3 emissions. Given the essential role of observational constraints, it is necessary to discuss in detail the influence of different satellite observations on seasonal variations of inversion results. It is good to see that "observations from AIRS, IASI, and CrIS" will be included in future studies. I suggest authors, at least in current state, to compare the seasonal cycle of NH3 columns measured by all the satellite sensors and to discuss the potential influences of using different data. Besides, it is not clear what means the offline NHx simulation for the iterative adjoint inversions. Please clarify it.

2. Bottom-up method. There have been several recent studies that use bottom-up method to establish high resolution emission inventory for NH3 in China. Most of these inventories peak its emissions during summer months, as shown in the literature review part of this paper. Therefore, in my opinion, improving NH3 inventory with strong seasonal cycle is not completely novel. The paper readers may ask what are the improvements and new points of this study in terms of approaches taken with the inventory development. These concerns are suggested to be clearly clarified in the revised manuscript.

3. Results. I think the paper would be stronger if the improved emission inventory is compared in detail with previous bottom-up inventories. Table 1 presents comparison of national emission totals. Because this paper shows more concern on seasonal variations of NH3 emissions, more comparisons are needed to evaluate the new emission inventory, especially for seasonal variability.

4. Evaluation. I have significant concerns about the emission inventory evaluation with surface measurement of NHx. If I understand correctly, the GEOS-Chem model results for 2008 are directly compared against measurement data over 2008-2012 period. If this is the case, it may involve large uncertainties due to varying meteorological conditions and varying concentrations of SO2, NOx and oxidants in the atmosphere from year to year. It would be better to conduct an air quality modeling for 2008-2012.

The NH3 emission used for the 5 years of model simulations can be fixed at 2008 because of small interannual variations. Or if the authors would not like to do this time-consuming work, I suggest only the measurement data for the year of 2008 can be used for model evaluation.
* * *

---

## Referee Comment (RC3) · Anonymous Referee #3 · 12 Oct 2017

This manuscript first derives top-down estimation of growing season NH3 emissions in China using TES satellite NH3 retrievals and GEOS-Chem adjoint model. Based on published methodology, it then develops an improved bottom-up NH3 inventory from fertilizer application and animal wastes in China. It finally applies both the top-down and bottom-up NH3 inventories in the GEOS-Chem forward model to compare with in situ surface measurements of ammonia and ammonium wet deposition, showing that both the inventories improve the model performance. The manuscript is well motived, scientifically sound, and well written. I recommend publication after the following comments are addressed.

First, there is a lack of detailed comparison between the top-down inventory and the bottom-up inventory developed by the authors (e.g. spatial, seasonal), as well as a

lack of discussion if the improved bottom-up inventory would match better with the TES retrieval. They showed that both inventories improved model simulation of surface wet deposition fluxes of ammonium, but this is indirect evidence and hard to interpret with regards to the emissions effect.

Second, it would help future studies if the bottom-up inventory developed by this study can be compared more quantitatively with the existing ones analyzed in the manuscript. A good place would be to plot that inventory in Figure 1 in comparison with the other ones displayed in the Figure.

Third, on line 109 they stated that the difference in bottom-up inventories is due to different base year, but in later places they stated that satellite data do not show a large trend of NH3 emissions in China and their model simulation was for the year of 2008 only, in spite of the use of multi years of TES observations. So my question is whether emissions would differ significantly by year, and if so, it would improve the scope of the manuscript if discussion could be added on the representativeness of year 2008 emissions they developed as the bottom-up inventory for other years, as well as offering suggestions on how scaling factors can be applied if their inventory is applied to other years.

Finally a technical issue about the GEOS-Chem model. It states that the model uses RPMARES as its thermodynamic module (line 172). I think the GEOS-Chem standard version uses ISORROPIA II thermodynamic equilibrium model. Is there a particular reason why the standard model setting is not used?
* * *

---

## Author Comment (AC2) · 24 Nov 2017

**Comment:** This manuscript uses both top-down and bottom-up methods to investigate the spatial and temporal variations of agricultural ammonia emissions in China. The top-down estimates of NH3 emissions, constrained by TES satellite NH3 observations and optimized by GEOS-Chem adjoint model, show a summer peak that is underestimated in current bottom-up emissions inventories. To resolve the seasonal difference, the authors construct a new bottom-up inventory that takes account of seasonal variability in fertilizer application rates and emissions factors. The improved bottom-up inventory is broadly consistent with the top-down inversion results; both are validated by surface concentrations of NH3 and wet deposition fluxes of NH4+. Overall, I think the paper reads well, provides interesting results and deserves publication. I include

some minor comments and suggested revisions in the following text.

**Response:** We thank the reviewer for the valuable comments. All of them have been addressed in the revised manuscript. Please see our itemized responses below.

**Comment:** 1. Inverse method. The TES satellite NH3 columns are included in the observation vector, and these measurements are the basis for deriving seasonal variations in inverted NH3 emissions. Given the essential role of observational constraints, it is necessary to discuss in detail the influence of different satellite observations on seasonal variations of inversion results. It is good to see that "observations from AIRS, IASI, and CrIS" will be included in future studies. I suggest authors, at least in current state, to compare the seasonal cycle of NH3 columns measured by all the satellite sensors and to discuss the potential influences of using different data. Besides, it is not clear what means the offline NHx simulation for the iterative adjoint inversions. Please clarify it.

**Response:** We thank the reviewer for the suggestion. TES NH3 measurement is the only satellite dataset available to us when the study was conducted. We also think a comparison of different satellite retrievals from TES, AIRS, IASI, and CrIS will help us better understand the spatial and seasonal patterns of NH3 over China. This requires a deep analysis of different satellite datasets (with retrieval vertical sensitivity, i.e. averaging kernel matrix)). We think it is beyond the scope of this paper and should be a separated study.

As for the offline NHx simulation, we now state in the text "To lower the computational expenses, we follow the approach of Paulot et al. (2014) and use an offline NHx (NH3 + NH4+) simulation for the adjoint inversion that only calculates the physical and chemical transformation of NHx driven by hourly simulated sulfate and total nitrate (HNO3 + NO3−) concentrations archived from the standard simulation".

**Comment:** 2. Bottom-up method. There have been several recent studies that
use bottom-up method to establish high resolution emission inventory for NH3 in China. Most of these inventories peak its emissions during summer months, as shown in the literature review part of this paper. Therefore, in my opinion, improving NH3 inventory with strong seasonal cycle is not completely novel. The paper readers may ask what are the improvements and new points of this study in terms of approaches taken with the inventory development. These concerns are suggested to be clearly clarified in the revised manuscript.

**Response:** Based on our review in Sect. 2, the commonly used Chinese NH3 emissions are rather inconsistent with respect to the summer peak. We now state in Sect. 2: "Huang et al. (2012) suggests a weak summer peak in Chinese NH3 emissions, while the MASAGE inventory (Paulot et al., 2014) indicates largest emissions in April and July. NH3 emission estimates of Streets et al. (2003) have a strong peak in June, and are much higher than Huang et al. (2012) and Paulot et al. (2014) in winter". That is why we need to better constrain the Chinese NH3 emissions using both top-down and bottom-up approaches. Out bottom-up emission inventory as we state in the abstract and in the manuscript that "includes more detailed information on crop-specific fertilizer application practices and better accounts for meteorological modulation of NH3 emission factors in China".

The revised manuscript also includes more information on comparison of our bottom-up emission inventory with previous estimates as described in the response below.

**Comment:** 3. Results. I think the paper would be stronger if the improved emission inventory is compared in detail with previous bottom-up inventories. Table 1 presents comparison of national emission totals. Because this paper shows more concern on seasonal variations of NH3 emissions, more comparisons are needed to evaluate the new emission inventory, especially for seasonal variability.

**Response:** We have now added in Figure 1 our improved bottom-up NH3 emission inventory. We also state in Sect. 5.3: "The spatial distribution and seasonal variations of our bottom-up NH3 emission inventory are also presented in Figure 1 for comparison with previous estimates. We can see that our bottom-up estimates show similar spatial features compared with Huang et al. (2012) and REAS v2, but also with some differences regionally. The total anthropogenic emission estimate of 11.7 Tg a−1 is in the middle of previous bottom-up estimates as summarized in Table 1, however, our NH3 emissions show much more distinct seasonal variations than previous estimates (e.g., Streets et al. (2003)) with emissions a factor of 3 higher in summer than winter."

**Comment:** 4. Evaluation. I have significant concerns about the emission inventory evaluation with surface measurement of NHx. If I understand correctly, the GEOS-Chem model results for 2008 are directly compared against measurement data over 2008-2012 period. If this is the case, it may involve large uncertainties due to varying meteorological conditions and varying concentrations of SO2, NOx and oxidants in the atmosphere from year to year. It would be better to conduct an air quality modeling for 2008-2012. The NH3 emission used for the 5 years of model simulations can be fixed at 2008 because of small interannual variations. Or if the authors would not like to do this time-consuming work, I suggest only the measurement data for the year of 2008 can be used for model evaluation.

**Response:** Thank you for the suggestion. We have now conducted a GEOS-Chem model simulation over 2008–2012 with the improved bottom-up NH3 emissions for comparison against surface measurements. We fixed anthropogenic emissions of NOx and SO2 to the year 2008 conditions (due to a lack of relevant interannual variations in this model version) for testing the influences from varying meteorological conditions. We now state in Sect. 5.3: "To test the influences from varying meteorology, we have conducted a model simulation over 2008–2012 with our improved bottom-up NH3 emissions and other SO2 and NOx anthropogenic emissions fixed to the year 2008 conditions. Our results show small differences in simulated seasonal mean NH4+ wet deposition fluxes and NH3 gas concentrations between the 2008 and 5–year averaged model results except for the wintertime surface NH3 concentrations that the 2008 model results are 14% lower (Figs. 8 and 9 as discussed below vs. Supplemental Fig.

S2)".

---

## Author Response (AR1)

J. Collett (Referee)

**Comment:** Zhang et al have done a terrific job using models and observations to improve understanding of ammonia emissions in China. Not only do they do a top-down analysis, using the GEOS-CHEM adjoint constrained by TES column NH3 measurements, to improve the seasonal and spatial variability in NH3 emissions, they then do a very thorough job improving past bottom-up inventories through careful analysis of fertilization practices and animal emissions. Combined, these make for a very strong paper – one of the best I have reviewed in some time.

I recommend the authors attend to a few comments in revising the manuscript:

**Response: We thank Prof. Collett for the valuable comments. We have addressed all of them in the revised manuscript, and please see the itemized responses below.**

**Comment:** 1. One of the main challenges in accurately simulating ammonia concentrations in chemical transport models is the treatment of dry deposition. Considerable attention has been paid recently to including more realistic, bi-directional flux parameterizations and this seems to help quite a lot in some regional simulations. Without a bidirectional treatment, NH3 loss rates by dry deposition can be biased high. While I am OK with the authors not including a bidi treatment in their model simulations here, I do think they should add some discussion how its absence

might influence their results. This is relevant to the top-down NH3 emissions estimates and to the comparison of model vs. surface concentration and wet deposition estimates.

**Response: Thank you for pointing it out. We now add the following text in Sect. 5.3 to discuss the bi-directional NH$_3$ flux: "Furthermore, while land-atmosphere exchange of NH$_3$ is bi-directional, the model here treats it as one-way emission and dry deposition processes. Zhu et al. (2015) previously implemented a bi-directional NH$_3$ exchange algorithm in GEOS-Chem, and they found that it led to small changes in wet deposition fluxes but had large impacts on emission estimates and surface concentrations over eastern China. Future work is required to improve the bi-directional exchange processes in the model."**

**Added Reference: Zhu, L., Henze, D., Bash, J., Jeong, G.-R., Cady-Pereira, K., Shephard, M., Luo, M., Paulot, F., and Capps, S.: Global evaluation of ammonia bidirectional exchange and livestock diurnal variation schemes, Atmos. Chem. Phys., 15, 12823-12843, https://doi.org/10.5194/acp-15-12823-2015, 2015.**

**Comment:** 2. Line 77: I suggest changing "together contribute" to "together are estimated to contribute"
**Response: changed as suggested.**

**Comment:** 3. Lines 150-157: the authors should discuss the Streets emission inventory here in the text. It is included in the figure and shows the strongest seasonality.
**Response: We now plot the spatial distribution of NH$_3$ emissions from the Streets inventory on Figure 1. We also state here: "NH$_3$ emission estimates of Streets et al. (2003) have a strong peak in June, and are much higher than Huang et al. (2012) and Paulot et al. (2014) in winter."**

**Comment:** 4. Line 173: I suggest changing "NH3 prefers to combine" to "NH3 is thermodynamically favored to combine"
**Response: changed as suggested.**

**Comment:** 5. Line 182: change "mixed clouds" to "mixed-phase clouds"

**Response: changed as suggested.**

**Comment:** 6. Lines 182-184: please explain and justify the retention efficiencies chosen for mixed-phase and cold clouds

**Response: Here we add the reference: "Wang, J., Hoffmann, A. A., Park, R. J., Jacob, D. J., and Martin, S. T.: Global distribution of solid and aqueous sulfate aerosols: Effect of the hysteresis of particle phase transitions, J. Geophys. Res., 113, D11206, doi:10.1029/2007jd009367, 2008".**

**Comment:** 7. Lines 248-249: are NH3 concentrations possibly also higher here because there are fewer NOx and SO2 emissions to generate acids that tie NH3 up in aerosols?

**Response: Thanks for pointing it out. We now state here "High $NH_3$ concentrations are also observed over Xinjiang province in Northwest China, which are likely emitted from animal grazing and remain mainly in gas phase due to lower $NO_x$ and $SO_2$ emissions to generate acids there."**

**Comment:** 8. Lines 336-338: How accurate/representative for China are the authors' assumptions here re: frequency of application of injection and broadcast fertilization methods?

**Response: We now state here "Based on fertilizer application practices, the first fertilizer application at plant is typically through injection (for rice and tobacco the first applications at both seeding and transplanting fields), and the rest by broadcast."**

**Comment:** 9. While the manuscript is generally quite well written, there are several small grammatical errors that should be corrected. The most significant are

a. Line 52: change "have" to "has" and "cause" to "causes"

b. Line 66: change "in the eastern China" to "in eastern China"

c. Line 129: change "human" to "humans"

d. Line 268: change "while overestimate" to "while they overestimate"

e. Line 276: change "increases in" to "increases are noted in"

f. Line 319: change "need to consider" to "requires considering"

g. Line 394: change "spending" to "spent"

h. Line 400: change "while only" to "while we only"

i. Line 442: change "needs to address" to "requires addressing"

j. Line 443: change "layer centered" to "layer is centered"

**Response: thank you for pointing them out. We have corrected them and a few other errors in the manuscript.**
* * *
**Reviewer 2**

**Comment:** This manuscript uses both top-down and bottom-up methods to investigate the spatial and temporal variations of agricultural ammonia emissions in China. The top-down estimates of NH3 emissions, constrained by TES satellite NH3 observations and optimized by GEOS-Chem adjoint model, show a summer peak that is underestimated in current bottom-up emissions inventories. To resolve the seasonal difference, the authors construct a new bottom-up inventory that takes account of seasonal variability in fertilizer application rates and emissions factors. The improved bottom-up inventory is broadly consistent with the top-down inversion results; both are validated by surface concentrations of NH3 and wet deposition fluxes of NH4+. Overall, I think the paper reads well, provides interesting results and deserves publication. I include some minor comments and suggested revisions in the following text.

**Response: We thank the reviewer for the valuable comments. All of them have been addressed in the revised manuscript. Please see our itemized responses below.**

**Comment:** 1. Inverse method. The TES satellite NH3 columns are included in the observation vector, and these measurements are the basis for deriving seasonal variations in inverted NH3 emissions. Given the essential role of observational constraints, it is necessary to discuss in detail the influence of different satellite observations on seasonal variations of inversion results. It is good to see that "observations from AIRS, IASI, and CrIS" will be included in future studies. I suggest authors, at least in current state, to compare the seasonal cycle of NH3 columns measured by all the satellite sensors and to discuss the potential influences of using different data. Besides, it is not clear what means the offline NHx simulation for the iterative adjoint inversions. Please clarify it.

**Response: We thank the reviewer for the suggestion. TES NH₃ measurement is the only satellite dataset available to us when the study was conducted. We also think a comparison of different satellite retrievals from TES, AIRS, IASI, and CrIS will help us better understand the spatial and seasonal patterns of NH₃ over China. This requires a deep analysis of different satellite datasets (with retrieval vertical sensitivity, i.e. averaging kernel matrix)). We think it is beyond the scope of this paper and should be a separated study.**

**As for the offline NH$_x$ simulation, we now state in the text "To lower the computational expenses, we follow the approach of Paulot et al. (2014) and use an offline NH$_x$ (NH$_3$ + NH$_4^+$) simulation for the adjoint inversion that only calculates the physical and chemical transformation of NH$_x$ driven by hourly simulated sulfate and total nitrate (HNO$_3$ + NO$_3^-$) concentrations archived from the standard simulation".**

**Comment:** 2. Bottom-up method. There have been several recent studies that use bottom-up method to establish high resolution emission inventory for NH3 in China. Most of these inventories peak its emissions during summer months, as shown in the literature review part of this paper. Therefore, in my opinion, improving NH3 inventory with strong seasonal cycle is not completely novel. The paper readers may ask what are the improvements and new points of this study in terms of approaches taken with the inventory development. These concerns are suggested to be clearly clarified in the revised manuscript.

**Response: Based on our review in Sect. 2, the commonly used Chinese NH₃ emissions are rather inconsistent with respect to the summer peak. We now state in Sect. 2: "Huang et al. (2012) suggests a weak summer peak in Chinese NH₃ emissions, while the MASAGE inventory (Paulot et al., 2014) indicates largest emissions in April and July. NH₃ emission estimates of Streets et al. (2003) have a strong peak in June, and are much higher than Huang et al. (2012) and Paulot et al. (2014) in winter". That is why we need to better constrain the Chinese NH₃ emissions using both top-down and bottom-up approaches. Out bottom-up emission inventory as we state in the abstract and in the manuscript that "includes more detailed information on crop-specific fertilizer application practices and better accounts for meteorological modulation of NH₃ emission**

**factors in China". The revised manuscript also includes more information on comparison of our bottom-up emission inventory with previous estimates as described in the response below.**

**Comment:** 3. Results. I think the paper would be stronger if the improved emission inventory is compared in detail with previous bottom-up inventories. Table 1 presents comparison of national emission totals. Because this paper shows more concern on seasonal variations of NH3 emissions, more comparisons are needed to evaluate the new emission inventory, especially for seasonal variability.

**Response: We have now added in Figure 1 our improved bottom-up NH$_3$ emission inventory. We also state in Sect. 5.3: "The spatial distribution and seasonal variations of our bottom-up NH$_3$ emission inventory are also presented in Figure 1 for comparison with previous estimates. The regional distribution of our bottom-up estimates is broadly similar to Huang et al. (2012) and REAS v2, but exhibits some important regional differences. The total anthropogenic emission estimate of 11.7 Tg a$^{-1}$ is in the middle of previous bottom-up estimates as summarized in Table 1, however, our NH$_3$ emissions show much more distinct seasonal variations than previous estimates (e.g., Streets et al. (2003)) with emissions a factor of 3 higher in summer than winter."**

**Comment:** 4. Evaluation. I have significant concerns about the emission inventory evaluation with surface measurement of NHx. If I understand correctly, the GEOS-Chem model results for 2008 are directly compared against measurement data over 2008-2012 period. If this is the case, it may involve large uncertainties due to varying meteorological conditions and varying concentrations of SO2, NOx and oxidants in the atmosphere from year to year. It would be better to conduct an air quality modeling for 2008-2012. The NH3 emission used for the 5 years of model simulations can be fixed at 2008 because of small interannual variations. Or if the authors would not like to do this time-consuming work, I suggest only the measurement data for the year of 2008 can be used for model evaluation.

**Response: Thank you for the suggestion. We have now conducted a GEOS-Chem model simulation over 2008–2012 with the improved bottom-up NH$_3$ emissions for comparison against surface measurements. We fixed anthropogenic**

emissions of $NO_x$ and $SO_2$ to the year 2008 conditions (due to a lack of relevant interannual variations in this model version) for testing the influences from varying meteorological conditions. We now state in Sect. 5.3: "To test the influences from varying meteorology, we have conducted a model simulation over 2008–2012 with our improved bottom-up $NH_3$ emissions and other $SO_2$ and $NO_x$ anthropogenic emissions fixed to the year 2008 conditions. Our results show small differences in simulated seasonal mean $NH_4^+$ wet deposition fluxes and $NH_3$ gas concentrations between the 2008 and 5–year averaged model results except for the wintertime surface $NH_3$ concentrations that the 2008 model results are 14% lower (Figs. 8 and 9 as discussed below vs. Supplemental Fig. S2)".
* * *
**Reviewer 3**

**Comment:** This manuscript first derives top-down estimation of growing season NH3 emissions in China using TES satellite NH3 retrievals and GEOS-Chem adjoint model. Based on published methodology, it then develops an improved bottom-up NH3 inventory from fertilizer application and animal wastes in China. It finally applies both the top-down and bottom-up NH3 inventories in the GEOS-Chem forward model to compare with in situ surface measurements of ammonia and ammonium wet deposition, showing that both the inventories improve the model performance. The manuscript is well motived, scientifically sound, and well written. I recommend publication after the following comments are addressed.

**Response: We thank the reviewer for the valuable comments. All of them have been addressed in the revised manuscript. Please see our itemized responses below.**

**Comment:** First, there is a lack of detailed comparison between the top-down inventory and the bottom-up inventory developed by the authors (e.g. spatial, seasonal), as well as a lack of discussion if the improved bottom-up inventory would match better with the TES retrieval. They showed that both inventories improved model simulation of surface wet deposition fluxes of ammonium, but this is indirect evidence and hard to interpret with regards to the emissions effect.

**Response: Thank you for the suggestion. We have compared model results using our bottom-up emission inventory with TES retrieved $NH_3$ columns. We now state in Sect. 5.3: "this strong seasonality is consistent with the adjoint optimized emission totals for March–October (50% higher in summer than spring and fall) considering uncertainties in the inversion results and satellite retrievals. The improved bottom-up Chinese $NH_3$ emissions are ~15% higher than the top-down estimates in May and June, and ~20% lower in other months. This can also be seen from the comparison of simulated $NH_3$ columns using the improved bottom-up inventory with the TES measurements (Supplemental Fig. S1)".**
**The revised manuscript also includes more information on comparison of our bottom-up emission inventory with previous estimates for the spatial and seasonal variations as described below.**

**Comment:** Second, it would help future studies if the bottom-up inventory developed by this study can be compared more quantitatively with the existing ones analyzed in the manuscript. A good place would be to plot that inventory in Figure 1 in comparison with the other ones displayed in the Figure.

**Response: Thank you for the suggestion. We have now plotted our bottom-up $NH_3$ emissions in Figure 1 for comparing with previous emission estimates. We add the following text in Sect. 5.3: "The spatial distribution and seasonal variations of our bottom-up $NH_3$ emission inventory are also presented in Figure 1 for comparison with previous estimates. The regional distribution of our bottom-up estimates is broadly similar to Huang et al. (2012) and REAS v2, but exhibits some important regional differences. The total anthropogenic emission estimate of 11.7 Tg a$^{-1}$ is in the middle of previous bottom-up estimates as summarized in Table 1, however, our $NH_3$ emissions show much more distinct seasonal variations than previous estimates (e.g., Streets et al. (2003)) with emissions a factor of 3 higher in summer than winter."**

**Comment:** Third, on line 109 they stated that the difference in bottom-up inventories is due to different base year, but in later places they stated that satellite data do not show a large trend of NH3 emissions in China and their model simulation was for the year of 2008 only, in spite of the use of multi years of TES observations. So my question is whether emissions would differ significantly by year, and if so, it would

improve the scope of the manuscript if discussion could be added on the representativeness of year 2008 emissions they developed as the bottom-up inventory for other years, as well as offering suggestions on how scaling factors can be applied if their inventory is applied to other years.

**Response: We stated on line 109 that "The factor of 2 difference is NOT likely due to the different base years". We also state "Analyses of historical NH$_3$ emissions in China show relatively stable or weak increasing trends (less than 3% per year) since 2000 (Xu et al., 2016; Kang et al., 2016), consistent with trends in atmospheric NH$_3$ concentration observed from satellites (Warner et al., 2017; Fu et al. 2017)". Thus we think that 2008 can be a representative year for the TES observational constraints. We are extending our bottom-up NH$_3$ emission inventory to other years and planning to report it in a separated study.**

**Comment:** Finally a technical issue about the GEOS-Chem model. It states that the model uses RPMARES as its thermodynamic module (line 172). I think the GEOS-Chem standard version uses ISORROPIA II thermodynamic equilibrium model. Is there a particular reason why the standard model setting is not used?

**Response: We now state here: "GEOS-Chem also includes the ISORROPIA II thermodynamic equilibrium model (Fountoukis and Nenes 2007). We find that differences in simulated monthly NH$_3$ concentrations over China by the two equilibrium models are less than 5%, and RPMARES runs about 30% faster in the GEOS-Chem adjoint."**

[revised manuscript text omitted]

strong peak in June, and are much higher than Huang et al. (2012) and Paulot et al. (2014) in winter. All
these discrepancies as discussed above emphasize the needs to improve our understanding of Chinese
$NH_3$ emissions in light of measurements of $NH_3$ gas concentration and deposition flux.

**3. Model description**

**3.1. The GEOS-Chem model**

Here we will use the GEOS-Chem CTM and its adjoint to simulate the sources and sinks of $NH_3$ over

China. GEOS-Chem is a global 3-D tropospheric chemistry model (http://geos-chem.org) driven by

assimilated meteorological data from the Goddard Earth Observing System (GEOS) of the NASA

Global Modeling and Assimilation Office (GMAO). The GEOS-5 meteorological data has a horizontal

resolution of 1/2° latitude × 2/3° longitude and a temporal resolution of 3 hours (1 hours for surface

variables). We apply here a one-way nested-grid version of GEOS-Chem with the native 1/2°×2/3°

horizontal resolution over East Asia (70°E–140°E, 15°N–55°N) and 2°×2.5° over the rest of the world

(Wang et al., 2004; Chen et al., 2009).

The model simulates a detailed tropospheric ozone–$NO_x$–hydrocarbon–aerosol chemistry as

described by Park et al. (2004) and Mao et al. (2010). $NH_3$ in the atmosphere is partitioned to gas and

aerosol phases based on the Regional Particulate Model Aerosol Reacting System (RPMARES)

thermodynamic equilibrium model (Binkowski and Roselle, 2003). $NH_3$ is thermodynamically favored

 to combine with $H_2SO_4$ to form ammonium bisulfate and ammonium sulfate, and excessive $NH_3$

can react with $HNO_3$ to form ammonium nitrate. GEOS-Chem simulations of secondary inorganic

aerosols (ammonium, sulfate, and nitrate) over China have been validated by Wang et al. (2013) and Li

et al. (2016) recently; both show high sensitivity of simulated nitrate concentrations to $NH_3$ emissions.

GEOS-Chem also includes the ISORROPIA II thermodynamic equilibrium model (Fountoukis and

Nenes 2007). We find that differences in simulated monthly $NH_3$ concentrations over China by the two

equilibrium models are less than 5%, and RPMARES runs about 30% faster in the GEOS-Chem

adjoint.

[revised manuscript text omitted]

Figure 2 shows TES observed NH$_3$ column concentrations with a footprint size of 5 km × 8 km from March to October. We do not analyze the late fall and winter months (November–February) as the valid TES observations become very limited, which hinders a reliable emission inversion in those months. As can be seen in Figure 2, the largest NH$_3$ column concentrations are observed over North

255 China reflecting intensive agricultural activities over this area. High NH$_3$ concentrations  are also observed over Xinjiang province in Northwest China, which are likely emitted from animal grazing and remain mainly in gas phase due to lower NO$_x$ and SO$_2$ emissions to generate acids there. 
[revised manuscript text omitted]

The spatial distribution and seasonal variations of our bottom-up $NH_3$ emission inventory are also presented in Figure 1 for comparison with previous estimates. The regional distribution of our bottom-up estimates is broadly similar to Huang et al. (2012) and REAS v2, but exhibits some important regional differences . The total anthropogenic emission estimate of 11.7 Tg $a^{-1}$ is in the middle of previous bottom-up estimates as summarized in Table 1, however, our $NH_3$ emissions show much more distinct seasonal variations than previous estimates (e.g., Streets et al. (2003)) with emissions a factor of 3 higher in summer than winter. As shown in Figure 7, this strong seasonality is consistent with the adjoint optimized emission totals for March–October (50% higher in summer than spring and fall) considering uncertainties in the inversion results and satellite retrievals. The improved bottom-up Chinese $NH_3$ emissions are ~15% higher than the top-down estimates in May and June, and ~20% lower in other months. This can also be seen from

the comparison of simulated $NH_3$ columns using the improved bottom-up inventory with the TES measurements (Supplemental Fig. S1).

For further and independent evaluation of these $NH_3$ emissions, we use an ensemble of surface measurements of $NH_3$ gas concentration and $NH_4^+$ wet deposition flux compiled by Zhao et al. (2017). The dataset includes monthly averages from a nationwide measurement network over China for 2011–2012 (Xu et al., 2015), ten sites in the North China Plain monitored by the Chinese Academy of Sciences for 2008–2010 (Pan et al., 2012), and two sites in China from EANET (2015). It should be noted that these measurements are compared with simulated results for 2008, inducing uncertainties from interannual variations. To test the influences from varying meteorology, we have conducted a model simulation over 2008–2012 with our improved bottom-up $NH_3$ emissions and other $SO_2$ and $NO_x$ anthropogenic emissions fixed to the year 2008 conditions. Our results show small differences in simulated seasonal mean $NH_4^+$ wet deposition fluxes and $NH_3$ gas concentrations between the 2008 and 5–year averaged model results except for the wintertime surface $NH_3$ concentrations that the 2008 model results are 14% lower (Figs. 8 and 9 as discussed below vs. Supplemental Fig. S2).

[revised manuscript text omitted]